# Crystal Structure, Electrical Conductivity and Hydration of the Novel Oxygen-Deficient Perovskite La$_2$ScZnO$_{5.5}$, Doped with MgO and CaO

**Ksenia Belova** [1,2,*] **, Anastasia Egorova** [1,2] **, Svetlana Pachina** [1] **and Irina Animitsa** [1,*]

1    The Institute of Natural Sciences and Mathematics, Ural Federal University, Mira st. 19,
620075 Yekaterinburg, Russia; oav-hn@yandex.ru (A.E.); biyvash@yandex.ru (S.P.)

2    Institute of High Temperature Electrochemistry of the Ural Branch of the Russian Academy of Sciences,
Akademicheskaya st. 20, 620066 Yekaterinburg, Russia

*    Correspondence: Ksenia.Belova@urfu.ru (K.B.); irina.animitsa@urfu.ru (I.A.)

**Abstract:** This paper demonstrates the possibility of creating oxygen deficiency in perovskites A$^{+3}$B$^{+3}$O$_3$ by introducing two types of cations with different charges into the B-sublattice. For this, it is proposed to introduce a two-charged cation, for example, Zn$^{2+}$, as an alternative to alkaline earth metals. Previously, this possibility was demonstrated for aluminate LaAlO$_3$ and indate LaInO$_3$. In this article, we have focused on the modification of the scandium-containing perovskite LaScO$_3$. The novel oxygen-deficient perovskite La$_2$ScZnO$_{5.5}$ and doped phases La$_{1.9}$Ca$_{0.1}$ScZnO$_{5.45}$, La$_2$Sc$_{0.9}$Ca$_{0.1}$ZnO$_{5.45}$, and La$_2$Sc$_{0.9}$Mg$_{0.1}$ZnO$_{5.45}$ were obtained via a solid-state reaction process. Their phase composition and hydration were investigated by XRD and TGA + MS techniques. The conductivities of these materials were measured by the electrochemical impedance technique under atmospheres of various water vapor partial pressures. All phases crystallized in orthorhombic symmetry with the *Pnma* space group. The phases were capable of reversible water uptake; the proton concentration increased in the order of La$_2$ScZnO$_{5.5}$ < La$_2$Sc$_{0.9}$Mg$_{0.1}$ZnO$_{5.45}$ < La$_2$Sc$_{0.9}$Ca$_{0.1}$ZnO$_{5.45}$ ≈ La$_{1.9}$Ca$_{0.1}$ScZnO$_{5.45}$ and reached ~90% hydration limit for Ca$^{2+}$-doped phases. The total conductivities increased with the increase in the free lattice volume in the sequence of σLa$_2$ScZnO$_{5.5}$ < σLa$_2$Sc$_{0.9}$Mg$_{0.1}$ZnO$_{5.45}$ < σLa$_{1.9}$Ca$_{0.1}$ScZnO$_{5.45}$ < σLa$_2$Sc$_{0.9}$Ca$_{0.1}$ZnO$_{5.45}$, the activation energy decreased in the same sequence. The sample La$_2$Sc$_{0.9}$Ca$_{0.1}$ZnO$_{5.45}$ showed the highest conductivity of about 10$^{-3}$ S·cm$^{-1}$ at 650 °C (dry air pH$_2$O = 3.5·10$^{-5}$ atm). Water incorporation was accompanied by an increase in conductivity in wet air (pH$_2$O = 2·10$^{-2}$ atm) due to the appearance of proton conductivity. The sample La$_2$Sc$_{0.9}$Ca$_{0.1}$ZnO$_{5.45}$ showed a conductivity of about 10$^{-5}$ S·cm$^{-1}$ at 350 °C (pH$_2$O = 2·10$^{-2}$ atm). A comparison of conductivities of obtained phase La$_2$ScZnO$_{5.5}$ with the conductivities of La$_2$AlZnO$_{5.5}$ and La$_2$InZnO$_{5.5}$ was made; the nature of the B-cation did not significantly affect the total conductivity.

**Keywords:** perovskite; acceptor doping; conductivity; water uptake; proton transport

## 1. Introduction

Proton and oxygen-ion conductors are studied intensively because of their potential applications as electrolytes, membranes, sensors, etc. [1,2]. Currently, the usage of such materials in solid oxide fuel cells (SOFC) is of special interest [3,4]. SOFCs are electrochemical devices that convert the chemical energy of a reaction between a fuel and an oxidizer directly into electricity and heat. The usage of H$_2$ as an energy carrier in SOFCs is a more effective and more environmentally friendly way than the traditional burning of carbon-containing fuel. Solid oxide fuel cells can also operate as an electrolyzer for water splitting and H$_2$ production via electrical power. In such an operating mode, the SOFC functions as a solid oxide electrolysis cell (SOEC). Due to their ability to use and produce hydrogen, SOFCs are expected to play an important role in the transition to hydrogen energetics [5].

In order to improve the working parameters of SOFCs and to lower their operating temperature [6], materials science research is developing intensively; nevertheless, the compounds with perovskite or perovskite-related structures remain the most widely studied objects [7]. These compounds can exhibit high ion ($O^{2-}$ or $H^+$) conductivity since they can adapt to an oxygen-deficient $ABO_{3-\delta}$.

The presence of oxygen vacancies is a result of acceptor doping (charged vacancies $V_o^{\bullet\bullet}$) or the presence of coordinatively unsaturated polyhedra (non-charged structural vacancies $V_o^{\times}$). In general, a low acceptor doping level (due to the limited solubility of the dopant) allows the creation of low oxygen vacancy concentrations that are randomly distributed, e.g., $BaZr_{1-x}Y_xO_{3-\delta}$ [8]. Higher oxygen deficiency can be specified when organizing multi-sublattice structures (usually two sublattices of B and B′) with proper combinations of oxidation states, e.g., $Ba_4Ca_2Nb_2O_{11}$ [9]. In this case, oxygen-deficient perovskites with ordered (or disordered) structures can be formed. An alternative way to introduce disorder in perovskite derivatives is to change the ratio of octahedra or tetrahedra (or polyhedra with a coordination number of five), which implies the obtaining of block structures (coherent intergrowth structures), e.g., $Ba_5Er_2Al_2ZrO_{13}\equiv2Ba_2ErAlO_5\cdot BaZrO_3$ [10]. All these paths of creating oxygen deficiency are shown in Figure 1.

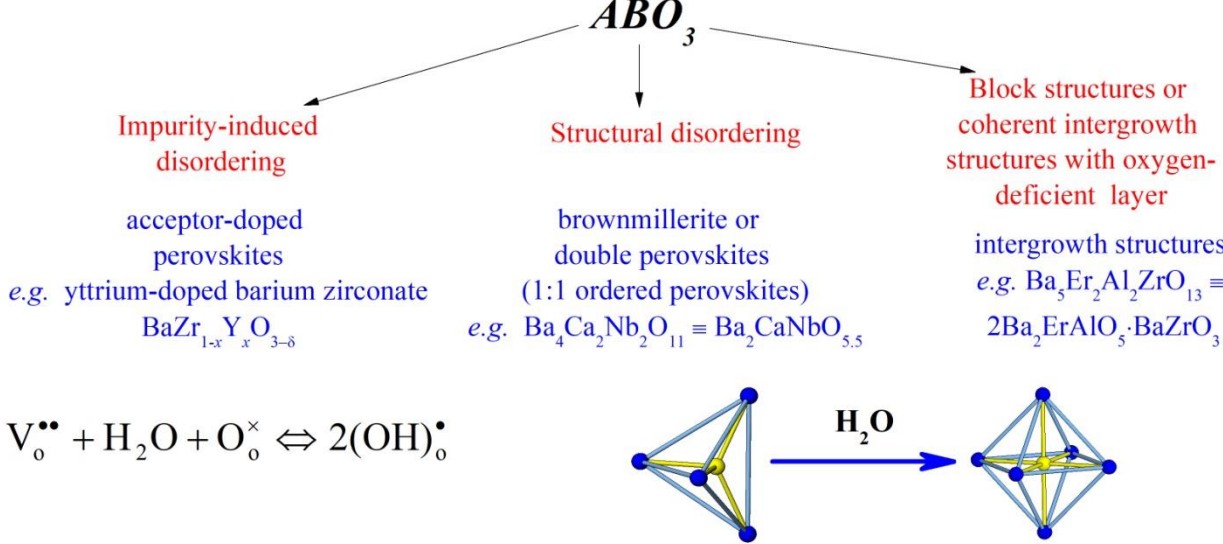

**Figure 1.** Main methods for modifying of perovskite structure to create an oxygen deficiency, the examples of basic materials and basic concepts of water incorporation.

As also shown in Figure 1, such phases are capable of incorporating water from the gas phase. For acceptor-doped perovskites, the process of the appearance of proton defects ($OH^-$ groups) is described in terms of the quasi-chemical formalism as the interaction between water vapor and oxygen vacancies $V_o^{\bullet\bullet}$. For the perovskites or perovskite-related phases in which atoms can adopt coordination numbers lower than six, the presence of such coordinatively unsaturated polyhedrons suggests the possibility of the dissociative water insertion from the gas phase. Upon hydration, tetrahedra are transformed to octahedrons, in which $OH^-$ groups participate in the coordination of cations. These phases can uptake a significant quantity of water and show a dominant proton transport below 500–600 °C.

Most research is focused on the investigations of compounds with acceptor doping; however, materials with structural oxygen vacancies have recently received considerable interest. Oxygen-deficient compounds such as $Ba_4Ca_2Nb_2O_{11}\equiv Ba_2CaNbO_{5.5}\equiv BaCa_{0.5}Nb_{0.5}O_{2.75}$, $Sr_4Sr_2Nb_2O_{11}$, $Sr_4Sr_2Ta_2O_{11}$ [9,11–17], $Ba_4Ca_2Ta_2O_{11}$ [18], $Ba_4Na_2W_2O_{11}$ [19], $Ba_2InSnO_{5.5}$ [20], $Ba_2YSnO_{5.5}$ [21] and $Ba_4In_2Zr_2O_{11}$ [22] are well known and well studied. Their structures have 8.33% vacant oxygen positions resulting from the combination of two types of cations in the B-sublattice in a 1:1 ratio. All of the aforementioned compounds are characterized

by predominant oxygen ion transport at high temperatures and proton transport at low temperatures and in a humid atmosphere. The distribution of cations in the lattice can be both statistical and ordered. The tendency for the appearance of a superstructure is associated with the differences between oxidation states of B-cations (and their radii): the higher the differences, the more order. For example, the structure of $Ba_4Na^{1+}_2W^{6+}_2O_{11}$ is ordered (with respect to B-cations) and described by the space group *Fm3m* with a doubled perovskite parameter and statistical arrangement of oxygen vacancies (double perovskite), but the compound $Ba_4In^{3+}_2Zr^{4+}_2O_{11}$ exhibits the space group *Pm3m* (ordinary perovskite). More recently, high ionic conductivity ($O^{2-}$, $H^+$) has been described for phases with a more complex structure, such as coherent intergrowth structures [10].

A new trend in materials science known as the "alkaline earth elements free strategy" is related to the development of perovskites based on the composition $A^{3+}B^{3+}O_3$. That is, the investigation of compounds not containing alkaline-earth components because the presence of an alkaline earth element in the composition of perovskite can result in the formation of the corresponding carbonates [23–26], and thus to the degradation of the material.

The well-studied orthorhombic-doped perovskite $LaScO_3$ exhibits high oxygen-ionic and proton conductivities. This complex oxide has been successfully doped into various sublattices [27–41]. In a dry atmosphere, acceptor-doped $LaScO_3$ is a mixed oxygen ion and hole conductor. In an atmosphere with high water vapor pressure, the total conductivity increases with the increase in water partial pressure due to the appearance of the proton contribution. Proton defects are a result of the incorporation of water molecules on oxygen vacancies and the formation of $OH^-$ groups. Of interest is the further modification of this phase by introducing a cation of a different nature into the B-sublattice at a 1:1 ratio to create oxygen deficiency. We decided to choose the double-charged zinc ion $Zn^{2+}$ as a second cation in the B-sublattice and to synthesize a new compound $La_2ScZnO_{5.5}$. Due to the stable oxidation state of zinc, the doping by $Zn^{2+}$ will not lead to an increase in the electronic conductivity; the introduction of zinc can also reduce the sintering temperature and help to obtain dense ceramics [42–44]. Moreover, zinc is not an alkaline earth element and therefore cannot impair chemical stability. It should also be emphasized that the substitution of $Sc^{3+}$ with $Zn^{2+}$ is a promising approach in view of the limited resources of scandium and its high cost.

$Zn^{2+}$ has already been used as a second B-cation for obtaining $Ba_2CeZnO_5$ [45]; moreover, the successful introduction of zinc into the B-sublattice of perovskite lanthanum aluminate and indate with the formation of the $LaAl_{0.5}Zn_{0.5}O_{2.75}$ and $LaIn_{0.5}Zn_{0.5}O_{2.75}$ has been shown [46,47]. In this work, different kinds of $Me^{2+}$ dopants were also used and $Me^{2+}$-substituted phases with compositions $La_{1.9}Ca_{0.1}ScZnO_{5.45}$ $La_2Sc_{0.9}Ca_{0.1}ZnO_{5.45}$, $La_2Sc_{0.9}Mg_{0.1}ZnO_{5.45}$ were synthesized. Magnesium and calcium were chosen as the dopants, due to their lower basicity and as a consequence lowest reactivity among alkaline earth metals. Here, we report crystal structure, conductivity, and water uptake for newly first-synthesized compounds.

## 2. Materials and Methods

### 2.1. Synthesis of Perovskite La$_2$ScZnO$_{5.5}$ and Doped Samples

The phase $La_2ScZnO_{5.5}$ and the doped samples $La_2Sc_{0.9}Mg_{0.1}ZnO_{5.45}$, $La_2Sc_{0.9}Ca_{0.1}ZnO_{5.45}$, and $La_{1.9}Ca_{0.1}ScZnO_{5.45}$ were synthesized via the solid-state method according to the following reactions:

$$La_2O_3 + ZnO + 0.50\,Sc_2O_3 \rightarrow La_2ScZnO_{5.5}, \tag{1}$$

$$La_2O_3 + ZnO + 0.45\,Sc_2O_3 + 0.10\,MgO(or\,CaO) \rightarrow La_2Sc_{0.9}Mg(Ca)_{0.1}ZnO_{5.45}, \tag{2}$$

$$0.45\,La_2O_3 + ZnO + 0.50\,Sc_2O_3 + 0.10\,CaO \rightarrow La_{1.9}Ca_{0.1}ScZnO_{5.45}. \tag{3}$$

High-purity powders (99.99% purity, REACHIM, Russian Federation) of $Sc_2O_3$, $La_2O_3$, CaO, and MgO were used as precursors. Before weighing, they were preliminarily calcinated: $Sc_2O_3$, CaO, and MgO were dried at 500 °C for 5 h to remove adsorption water;

$La_2O_3$ was calcinated at 1100 °C for 3 h to decompose the surface lanthanum carbonates $LaOHCO_3$ and $La_2O_2CO_3$ [48]. Stoichiometric amounts of oxides were weighed on an analytical balance (Sartorius AG, Göttingen, Germany) with an accuracy of $\pm 0.0001$ g, then mixed and manually ground with ethanol in an agate mortar and then calcinated in the temperature range of 800−1300 °C for 24 h with heating steps of 100 °C; intermediate grindings were made for every step.

## 2.2. X-ray Diffraction

Phase composition was monitored by powder X-ray diffraction (PXRD) on a D8 Advance diffractometer (Bruker, Billerica, MA, USA) with Cu K$\alpha$ radiation. The basic cell parameters of the samples were determined by Rietveld refinement using FULLPROF software [49]. The phases were identified in accordance with the Inorganic Crystal Structure Database (ICSD, version 2020/2).

## 2.3. Thermogravimetric Analysis Coupled with Mass Spectrometry

*Preparation of hydrated powder samples.* The samples were first treated at 1000 °C for 1 h in dry Ar to remove any volatile substances ($CO_2$, $H_2O$), then the samples were cooled at a rate of 1 °C/min in wet air (p$H_2O$ = $2 \cdot 10^{-2}$ atm) from which the $CO_2$ was removed.

Thermogravimetric analysis (TGA) was used to determine the proton concentrations. The TGA was performed on the preliminarily hydrated powder samples using an STA 409 PC analyzer (Netzsch, Selb, Germany) coupled with a quadrupole mass spectrometer QMS 403 C Aëolos (Netzsch, Selb, Germany).

## 2.4. Conductivity Measurements

*Preparation of compacted samples.* For electrical measurements, the powder samples were compacted to the pellets at 50 MPa on a PLG-12 laboratory hydraulic press (LabTools, Saint-Petersburg, Russian Federation). The solution of rubber in hexane was used as a plasticizer. The pellets were sintered at 1300 °C for 24 h. The diameters and thicknesses of the pellets were ~7 mm and 1–2 mm, respectively. The density of the compacted samples was determined by the Archimedes' method in kerosene, the preparation of the samples was carried out according to the standard procedure [50]. The values of relative densities were found to be ~95–97%. Then, the platinum paste electrodes were painted on polished surfaces of the sintered pellets and annealed at 900 °C for 3 h in air.

*Impedance spectroscopy.* The conductivity of the samples was characterized by an impedance spectroscopy technique. The *ac* conductivity of the samples was measured by the 2-probe method using the Z-1000P impedance spectrometer (Electrochemical Instruments (Elins), Chernogolovka, Russian Federation) within the frequency range of 1–$10^6$ Hz and amplitude of the perturbation signal 127 mV. The bulk resistance was calculated from a complex impedance plot using the Zview software fitting (Scribner Associates, Southern Pines, NC, USA). The accuracy of the impedance spectrometer was $\pm 3\%$. The measurement errors for a series of the samples did not exceed 10% for low temperatures (T < 500 °C) and 5% for high temperatures.

*Conditions of the measurements.* The measurements of the temperature dependencies of conductivities were performed in the temperature range of 200–1000 °C with a cooling rate of 1°/min and taken every 20 °C until a constant resistance value was established. The conductivity measurements were carried out in dry and wet atmospheres. To minimize the experimental error, measurements were carried out on pellets with different geometries (thicknesses); each composition was tested three times on different pellets.

*Preparation of "dry" and "wet" air.* In order to obtain the desired humidity and remove $CO_2$, the air was bubbled subsequently through 30% NaOH and saturated KBr solutions at room temperature to create "wet" air. For obtaining a "dry" atmosphere, the air was passed through $P_2O_5$ powders. Partial water vapor pressure was measured by the HIH-3610 humidity sensor (Honeywell, Freeport, USA) and was $2 \cdot 10^{-2}$ atm for "wet" and $3.5 \cdot 10^{-5}$ atm for "dry" air.

The main steps of the investigations are presented in Figure 2.

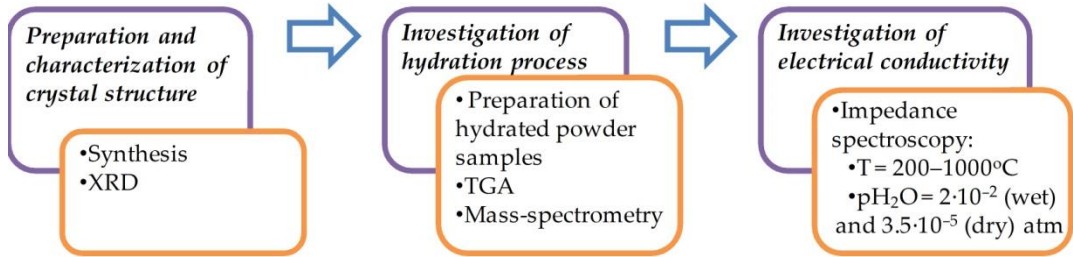

**Figure 2.** The scheme of investigations of perovskite $La_2ScZnO_{5.5}$ and doped samples.

## 3. Results and Discussion

### 3.1. Structure

Room temperature X-ray diffraction (XRD) patterns are shown in Figure 3. The new phase $La_2ScZnO_{5.5}$ has the same symmetry of the lattice as the orthorhombic perovskite $LaScO_3$ with the space group *Pnma*. Moreover, no additional superstructure lines were observed in the PXRD patterns, indicating a random distribution of cations in the B-sublattice without any ordering. All doped phases—$La_2Sc_{0.9}Mg_{0.1}ZnO_{5.45}$, $La_2Sc_{0.9}Ca_{0.1}ZnO_{5.45}$, and $La_{1.9}Ca_{0.1}ScZnO_{5.45}$—also crystallized in orthorhombic symmetry. The lattice parameters are shown in Table 1.

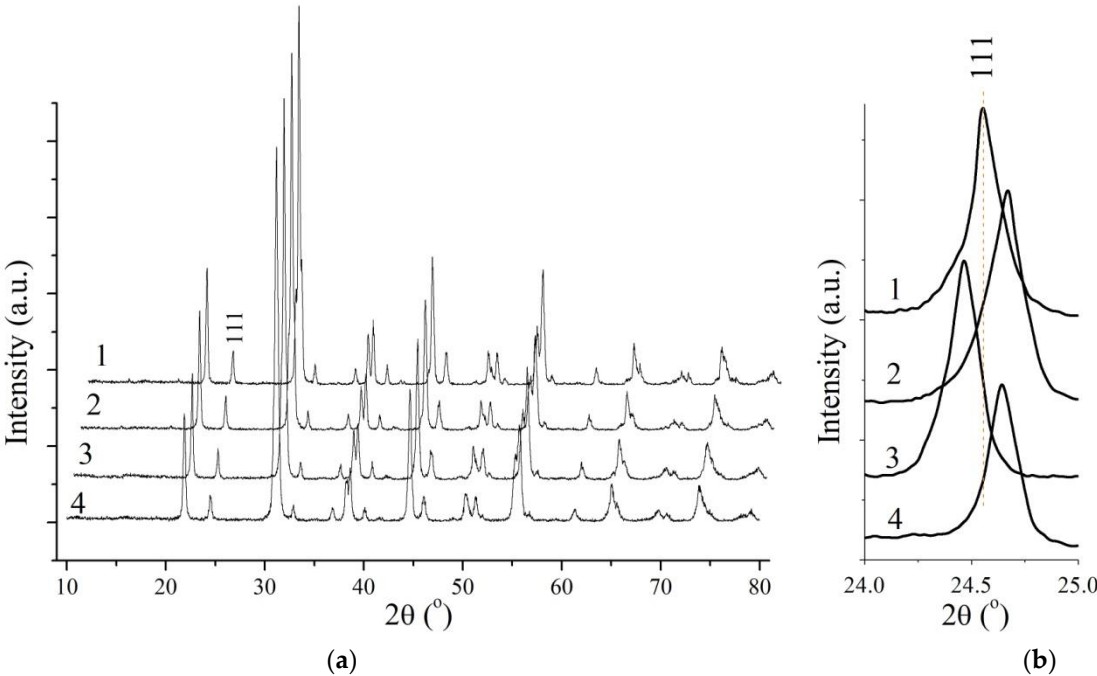

(a)          (b)

**Figure 3.** (**a**) The XRD patterns of new phase $La_2ScZnO_{5.5}$ (1) and doped phases $La_2Sc_{0.9}Mg_{0.1}ZnO_{5.45}$ (2), $La_2Sc_{0.9}Ca_{0.1}ZnO_{5.45}$ (3), $La_{1.9}Ca_{0.1}ScZnO_{5.45}$ (4); (**b**) shifts of the diffraction peak (111) for investigated phases.

The change in the lattice parameters correlates with the difference in the radii of the host ions and dopants. After doping with smaller ions, the unit cell parameters slightly decreased for $La_2Sc_{0.9}Mg_{0.1}ZnO_{5.45}$ and $La_{1.9}Ca_{0.1}ScZnO_{5.45}$ ($rLa^{3+}$ = 1.36 Å, $rCa^{2+}$ = 1.34 Å; $rSc^{3+}$ = 0.745 Å, $rMg^{2+}$ = 0.72 Å [51] for corresponding coordination numbers). At the same time, for the composition, the $La_2Sc_{0.9}Ca_{0.1}ZnO_{5.45}$ $Sc^{3+}$ ion was replaced by a significantly larger ion $Ca^{2+}$ (r = 1.00 Å), which led to the expansion of the lattice. This is also confirmed by the shifts of the diffraction peaks; a shift to smaller angles represents the expansion of

the cell, and a shift towards larger angles corresponds to the contraction of the unit cell. Figure 3b shows this dependence using the diffraction peak as an example (111).

The lattice-free volume for the doped samples increased in the sequence $La_2ScZnO_{5.5} - La_2Sc_{0.9}Mg_{0.1}ZnO_{5.45} - La_{1.9}Ca_{0.1}ScZnO_{5.45} - La_2Sc_{0.9}Ca_{0.1}ZnO_{5.45}$ (Table 1).

**Table 1.** The lattice parameters for $La_2ScZnO_{5.5}$ and doped phases in comparison with the literature data for $La_2InZnO_{5.5}$ [47] and $La_2AlZnO_{5.5}$ [46].

| Formula | $a$, Å | $b$, Å | $c$, Å | $V$ *, Å³ | $V$ ** $_{free}$, Å³ |
|---|---|---|---|---|---|
| $La_2ScZnO_{5.5}$ | 5.796 (5) | 8.101 (9) | 5.683 (4) | 66.73 | 26.15 |
| $La_2Sc_{0.9}Mg_{0.1}ZnO_{5.45}$ | 5.792 (7) | 8.095 (7) | 5.678 (9) | 66.58 | 26.27 |
| $La_2Sc_{0.9}Ca_{0.1}ZnO_{5.45}$ | 5.813 (3) | 8.116 (2) | 5.685 (7) | 67.06 | 26.63 |
| $La_{1.9}Ca_{0.1}ScZnO_{5.45}$ | 5.794 (7) | 8.099 (8) | 5.681 (7) | 66.67 | 26.38 |
| $La_2InZnO_{5.5}$ [47] | 5.941 (2) | 8.217 (1) | 5.723 (7) | 69.86 | 29.08 |
| $La_2AlZnO_{5.5}$ [46] | 3.7932 (1) | − | − | 54.58 | 14.55 |

* Unit cell volume divided by Z. ** Lattice free volume is defined as the volume of the ions in the formula unit (Z = 4) subtracted from the unit cell volume divided by Z, where Z is a number of formula units.

### 3.2. Hydration Behavior

Thermogravimetric analysis was performed for the preliminarily hydrated $La_2ScZnO_{5.5}$ and all doped samples. The TG curves for all investigated samples are shown in Figure 4. All TG curves are characterized by mass loss. The mass spectrometry analysis confirmed that the mass changes can be attributed to water removal and no other volatile components (such as $CO_2$ or $O_2$) were detected. Figure 4 also shows the MS curve for hydrated sample $La_{1.9}Ca_{0.1}ScZnO_{5.45}$, as a typical example. The peaks of water removal on the MS curve corresponded to the stages of mass loss on the TG curve. According to the TG data for doped samples, the main mass loss occurred in the temperature range of 300–400 °C and the second stage of mass loss took place at 500–550 °C. The temperature ranges of these dehydration steps were the same for all doped samples. At the same time, the undoped sample $La_2ScZnO_{5.5}$ exhibited one effect of mass loss at the temperature of approximately 500–550 °C. For all samples, water was completely removed at T > 600 °C.

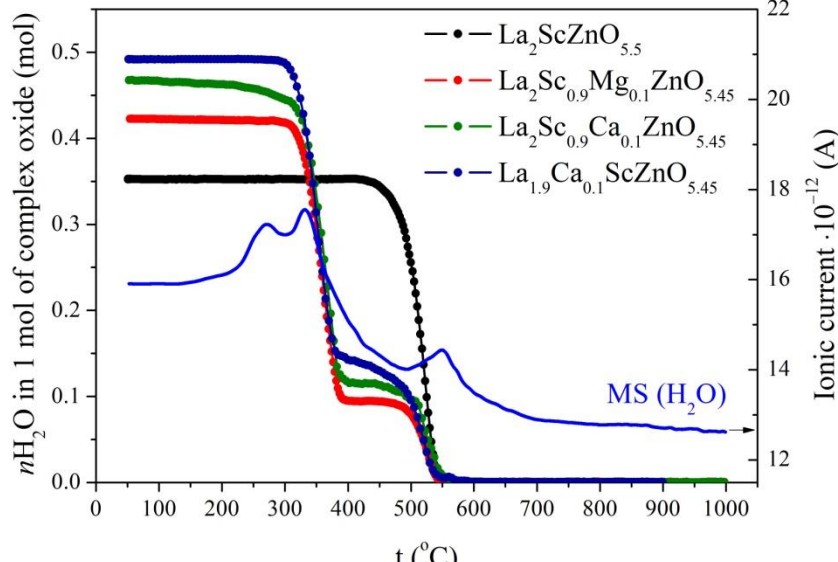

**Figure 4.** TG curves for hydrated $La_2ScZnO_{5.5}$, $La_2Sc_{0.9}Mg_{0.1}ZnO_{5.45}$, $La_2Sc_{0.9}Ca_{0.1}ZnO_{5.45}$, $La_{1.9}Ca_{0.1}ScZnO_{5.45}$ and MS-curve for $La_{1.9}Ca_{0.1}ScZnO_{5.45}$.

These results show that the states of the $OH^-$ groups in hydrated doped and undoped samples are different; therefore, different types of protonic species with different thermal stability are present in doped phases. The introduction of the dopants leads to the appearance of a part of $OH^-$ groups with a lower thermal stability. A possible cause may be different types of oxygen vacancies.

It is well known that protons can dissolve in oxygen-deficient phases through contact with water vapor. The water insertion into the structure of oxides can be represented as a process of dissociative dissolution of water with the formation of proton defects ($OH^-$ groups) and the reaction can be written by the Kröger–Vink notation as follows [52]:

$$H_2O + O_o^\times + V_o^\times \rightarrow (OH)'_o + (OH)_O^\bullet, \tag{4}$$

where $V_o^\times$ is the structural oxygen vacancy, $O_o^\times$ is the oxygen on regular sites, $(OH)_O^\bullet$ is the hydroxide group (a proton incorporated into the electron cloud of a regular oxygen atom) and $(OH)'_o$ is the hydroxide group in the site of a structural oxygen vacancy.

This equation is used for structural oxygen vacancies, i.e., in the case of oxygen deficiency is created by placing two cations with different charges into the B-sublattice. That is, for the undoped phase $La_2ScZnO_{5.5}$ $[V_o^\times]_{0.5}$ with structural oxygen vacancies, the hydration process can be recorded as well.

Thus, when oxygen vacancies are completely occupied by the oxygen from $H_2O$ molecules, the hydration limit can be reached. For the undoped sample $La_2ScZnO_{5.5}$, the observed water uptake was 0.35 mol $H_2O$, which is lower than the theoretical value of 0.50 mol $H_2O$. That is, the observed water uptake was about 70% of the concentration of oxygen vacancies, which means that not all oxygen vacancies are involved in hydration. This situation is typical for many perovskites. For example, the concentration of protons was measured in oxides based on $La_2O_3$: $La_{0.9}Sr_{0.1}MO_{3-\delta}$ (M = Al, Sc, In, Yb, Y) [53], $La_{0.9}Sr_{0.1}$ $(Yb_{1-x}M_x)O_{3-\delta}$ (M = Y, In) [54], $La_{0.9}M_{0.1}YbO_{3-\delta}$ (M = Ba, Sr, Ca, Mg) [55], $La_{1-x}Ba_xYbO_{3-\delta}$ [56], $La_{1-x}Ca_xScO_{3-\alpha}$ [33], and $La_{1-x}Sr_xScO_{3-\delta}$ [34]. The authors point out that the concentration of protons does not reach the concentration of oxygen vacancies. Possible reasons are changes in the local structure and the existence of non-equivalent oxygen atoms, trapping oxygen vacancies by acceptor defects. Comparing the obtained TG data for the synthesized phase $La_2ScZnO_{5.5}$ to the data on the hydration of doped scandates $LaScO_3$, we can conclude that the dehydration of scandates occurs at higher temperatures (T~1000 °C) and they have significantly lower proton concentrations.

All $Me^{2+}$-doped hydrated samples showed approximately the same values of water uptake (~0.48 mol $H_2O$ for $Ca^{2+}$-doped samples and 0.42 mol $H_2O$ for $Mg^{2+}$-doped sample) and these values are higher than those for undoped $La_2ScZnO_{5.5}$ (Figure 4). That is, the observed water uptake was about 90% and 75% of the concentration of oxygen vacancies (for $Ca^{2+}$ and $Mg^{2+}$-doped samples, accordingly). The doped phases have an additional amount of oxygen vacancies ($V_o^{\bullet\bullet}$), which appear in accordance with the equations:

$$2CaO \xrightarrow{La_2O_3} 2Ca'_{La} + 2O_o^\times + V_o^{\bullet\bullet}, \tag{5}$$

$$2MgO \xrightarrow{Sc_2O_3} 2Mg'_{Sc} + 2O_o^\times + V_o^{\bullet\bullet}, \tag{6}$$

$$2CaO \xrightarrow{Sc_2O_3} 2Ca'_{Sc} + 2O_o^\times + V_o^{\bullet\bullet}. \tag{7}$$

where $V_o^{\bullet\bullet}$ is the oxygen vacancy, $O_o^\times$ is the oxygen atom in a regular position, $Ca'_{La}$ is calcium in the lanthanum sublattice, $Ca'_{Sc}$ or $Mg'_{Sc}$ is calcium or magnesium in the scandium sublattice, respectively.

These vacancies interact with water vapor according to the following hydration equation:

$$V_o^{\bullet\bullet} + H_2O + O_o^\times \Leftrightarrow 2(OH)_o^\bullet, \tag{8}$$

where $V_o^{\bullet\bullet}$ is oxygen vacancy, $O_o^{\times}$ is the oxygen atom in a regular position, and $(OH)_o^{\bullet}$ is the hydroxyl group in the oxygen sublattice. On the one hand, this can explain an increase in the degree of hydration of doped phases, but a change in the local structure upon the introduction of the dopants can also lead to an increase in the availability of oxygen vacancies for hydration. A change in the local structure and, consequently, the appearance of different oxygen positions leads to a change in the size of oxygen vacancies (which are usually smaller than oxygen ions [57,58]) and the inaccessibility of some vacancies for replacement by OH⁻ groups. Therefore, an incomplete hydration occurs. Therefore, local distortions can cause different degrees of hydration for the samples with the same structural type.

The presence of various types of oxygen vacancies in doped $La_2ScZnO_{5.5}$ can explain the differences in temperature behavior of dehydration of the doped and undoped phases.

### 3.3. Conductivity

### 3.3.1. Analysis of Impedance Spectra

In Figure 5, the impedance spectra of the sample $La_{1.9}Ca_{0.1}ScZnO_{5.45}$ are shown in a Nyquist plot at several temperatures (a) and different air humidities (b), as an example. In general, we can distinguish the contributions due to bulk, grain boundary, and electrode effects. The first semicircle, starting from zero of the complex impedance plot, corresponds to the bulk resistance $R_b$ (the value of capacitance was in the range of $10^{-11}$–$10^{-10}$ F), the second one to the grain-boundary resistance $R_{gb}$ (the value of the capacitance was ~$10^{-9}$ F), and the third small semicircle in the low-frequency regions corresponds to the electrode contribution $R_{electrode}$ (the value of capacitance was ~$10^{-5}$ F). This shape of the spectra was typical for all investigated phases.

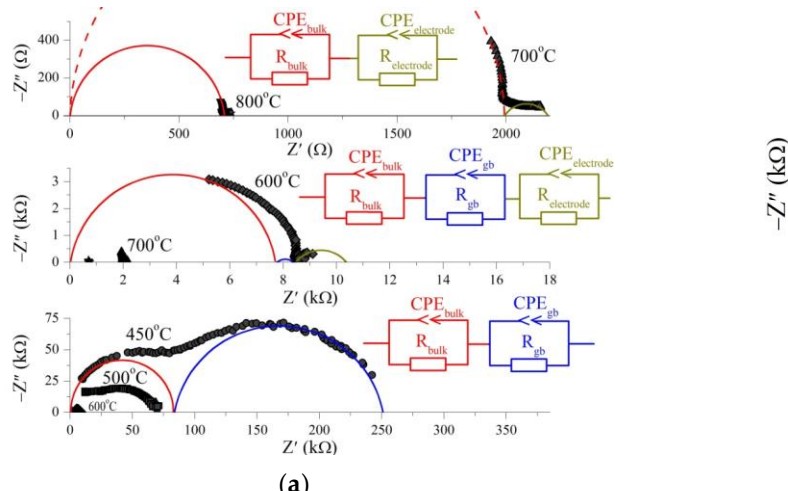
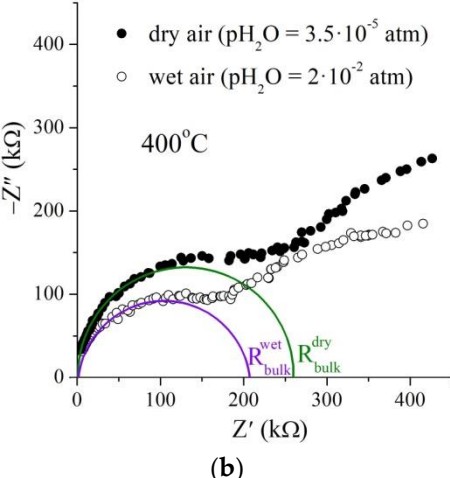

| (a) | (b) |
|---|---|

**Figure 5.** The Nyquist plots for the sample $La_{1.9}Ca_{0.1}ScZnO_{5.45}$. (**a**) Evolution of the spectra with temperature in dry air (pH$_2$O = 3.5·$10^{-5}$ atm) and corresponding equivalent circuits; (**b**) evolution of the spectra with water vapor partial pressure pH$_2$O = 2·$10^{-2}$ atm (open signs) and pH$_2$O = 3.5·$10^{-5}$ atm (closed signs) at 400 °C.

However, these contributions as well as the shape of the spectra changed with the temperature: with an increase in the temperature, the contribution of the grain boundary decreases, and the electrode contribution appears. In the temperature range below 500 °C, two overlapping semicircles are observed from the bulk and grain boundary responses. The bulk resistance of the samples was determined from the interceptions of the first high-frequency arc with the real axis at low frequencies. However, with the increase in the temperature, the grain boundary contribution decreases simultaneously with the disappearance of the first arc; and at higher temperatures (T > 700 °C), the bulk resistance

was found as the left-hand intercept of the third semicircle with the real axis. The possible uncertainty due to different extrapolations was estimated as $\Delta(\log\sigma) = \pm 0.10$.

The comparison of impedance spectra in wet and dry air at 400 °C is presented in Figure 5b, and it can be seen that the bulk resistance decreases with the increasing partial water vapor pressure.

Further in the article, only the bulk conductivity will be discussed.

### 3.3.2. Temperature Dependencies of Conductivity
Dry Conditions

The temperature dependencies of conductivities of the newly synthesized compound $La_2ScZnO_{5.5}$ compared with those of $La_2AlZnO_{5.5}$ [46] and $La_2InZnO_{5.5}$ [47] are shown in Figure 6.

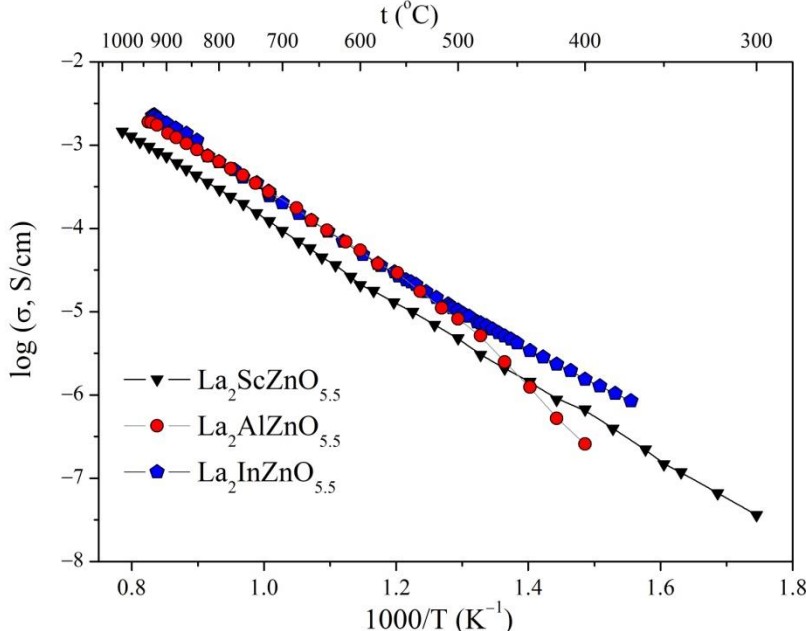

**Figure 6.** Temperature dependencies of the conductivities of $La_2ScZnO_{5.5}$, $La_2AlZnO_{5.5}$ [46], and $La_2InZnO_{5.5}$ [47].

As can be seen, the values of total conductivities are close to each other, although the $In^{3+}$-containing sample showed the highest conductivity; for the temperatures T < 500 °C, the electrical conductivity increases in the sequence $La_2AlZnO_{5.5} < La_2ScZnO_{5.5} < La_2InZnO_{5.5}$, and at high temperatures, the Sc sample had a conductivity by 0.3 orders of magnitude lower. All of these samples have the same number of oxygen ions and consequently oxygen deficiency (all elements are present in the highest stable oxidation states). Therefore, the changes in the conductivity may be due to the changes in the mobility of oxygen ions or due to the changes in electronic contribution. For understanding the phenomenon, studies on ionic transport numbers are needed, but in principle, the nature of the B-cation does not significantly affect the total conductivity.

The comparison of the temperature dependencies of conductivities of undoped phase $La_2ScZnO_{5.5}$ and $Ca^{2+}$, $Mg^{2+}$-doped phases is shown in Figure 7. It is clearly seen that doping increases conductivity. The increase in conductivity can be associated with an increase in the concentration of oxygen vacancies due to acceptor doping. However, doped samples are characterized by an equal level of oxygen deficiency, but the conductivities of these phases are significantly different. That is, the oxygen mobility in the doped phases is significantly different.

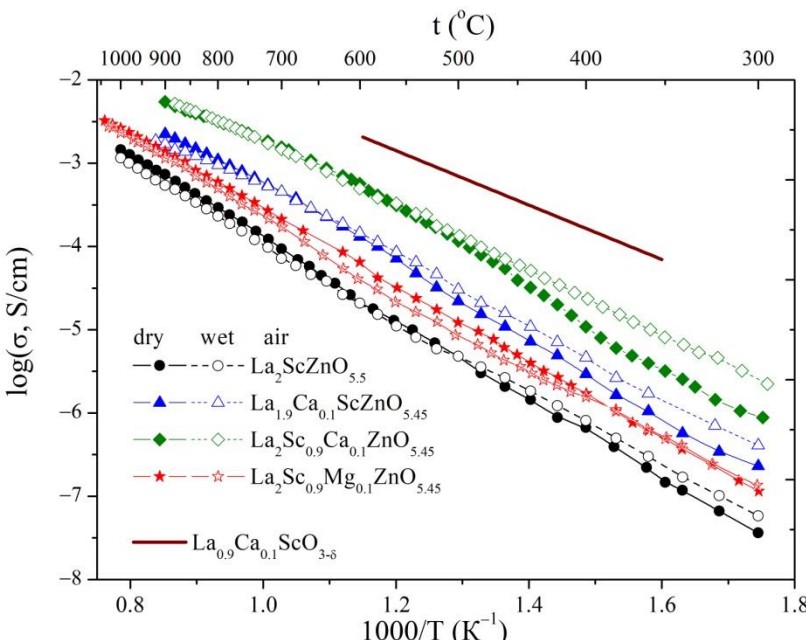

**Figure 7.** Temperature dependencies of the conductivities of $La_2ScZnO_{5.5}$, $La_2Sc_{0.9}Mg_{0.1}ZnO_{5.45}$, $La_2Sc_{0.9}Ca_{0.1}ZnO_{5.45}$, $La_{1.9}Ca_{0.1}ScZnO_{5.45}$ in comparison with $La_{0.9}Ca_{0.1}ScO_{3-\delta}$ [33].

In general, two main parameters are used to explain the changes in ion mobility—the lattice volume and the lattice-free volume. Usually, the $O^{2-}$ ionic conductivity increases with the increasing lattice volume as a result of the weakening metal–oxygen bonding, and, as a consequence, oxygen mobility increases. The lattice-free volume is defined as the difference between the lattice cell volume and the summarized volume occupied by all ions. As the lattice-free volume increases, the lattice spacing increases and the mobility of ions increases; therefore, ionic conductivity increases. Both of these factors can change symbatically with a change in the size of the ions or antibatically; however, it depends on the specific type of lattice. Therefore, both factors can be analyzed.

Figure 8 presents a comparison of the conductivities as a function of the lattice-free volume for all samples. As can be seen, upon doping, an increase in the lattice-free volume leads to the facilitation of oxygen transport and therefore to an increase in the total conductivity—a maximum conductivity is observed for the substitution $Ca^{2+} \rightarrow Sc^{3+}$. This is also confirmed by the calculated values of the activation energy (Figure 8). Activation energies can be calculated using the relation:

$$\sigma T = \sigma \exp(-E_a/kT), \qquad (9)$$

where T is the absolute temperature, $\sigma_o$ is the pre-exponential factor, $E_a$ is the activation energy, which was calculated from the slope of the $\log\sigma T$ versus $1000/T$ curve, and k is Boltzmann's constant.

From Figure 8, it can be derived that under substitution the increase in conductivity is accompanied by a decrease in activation energy. Moreover, it is well known that a conductivity increase and an activation energy decrease typically occur for oxygen ion-conducting materials. As it has been already shown in other works, a maximum conductivity is coupled with a minimum in the activation energy [59]. Therefore, the most favorable doping is $Ca^{2+}$ on the $Sc^{3+}$ sites.

Wet Conditions

The conductivity dependencies versus the reciprocal temperature in wet air are compared with those obtained in dry air in Figure 7. In a wet atmosphere below 500−550 °C, the conductivities of all samples are higher than those in dry air. Such behavior is a result of

the incorporation of water and the formation of proton current carriers below 500−550 °C. At high temperatures (T > 600 °C), water is removed (according to the TG data) and the conductivities in dry and wet atmospheres are comparable. Thus, the investigated phases are capable of exhibiting proton conductivity.

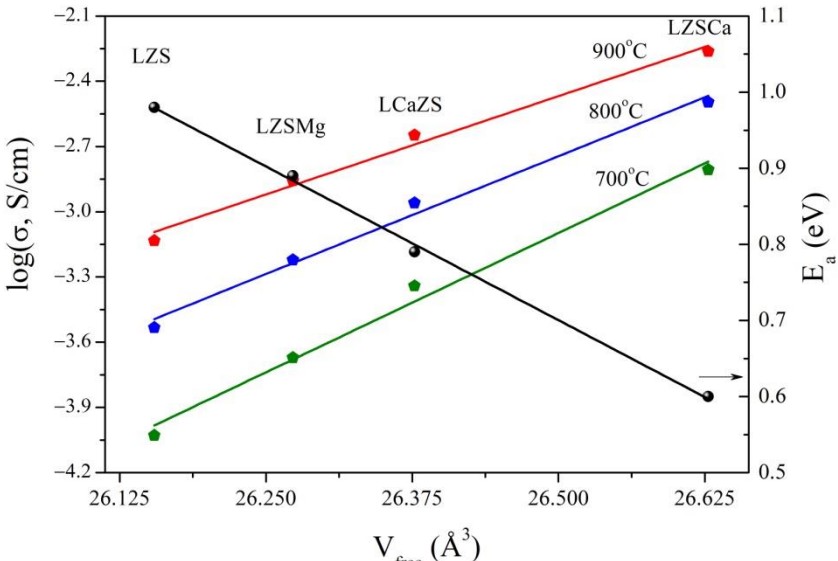

**Figure 8.** Comparison of the conductivities and activation energies as a function of the free lattice volume for high-temperature region (T > 700 °C) in dry air for the samples: $La_2ScZnO_{5.5}$ (LSZ), $La_2Sc_{0.9}Mg_{0.1}ZnO_{5.45}$ (LZSMg), $La_2Sc_{0.9}Ca_{0.1}ZnO_{5.45}$ (LZSCa), and $La_{1.9}Ca_{0.1}ScZnO_{5.45}$ (LCaZS).

A comparison of the conductivity in wet air shows similar trends as for dry air; it increases in the sequence: $La_2Sc_{0.9}Mg_{0.1}ZnO_{5.45} < La_{1.9}Ca_{0.1}ScZnO_{5.45} < La_2Sc_{0.9}Ca_{0.1}ZnO_{5.45}$. Figure 9 shows that conductivity increases with an increase in the lattice-free volume.

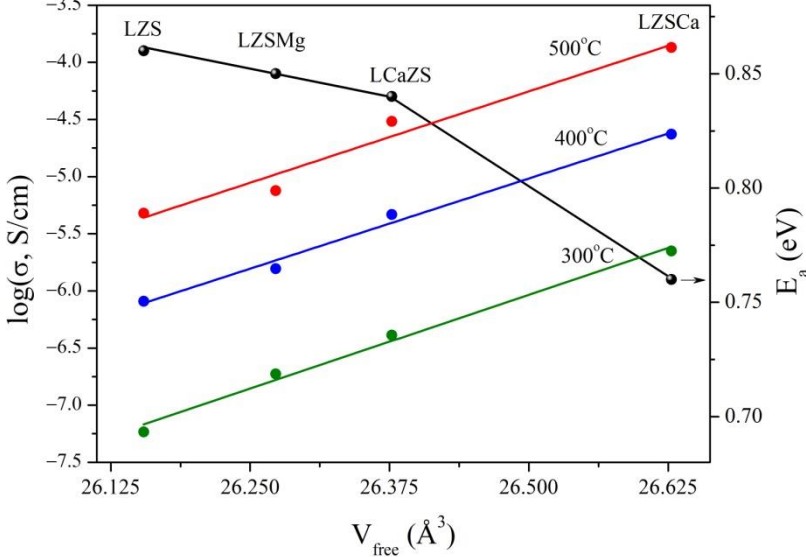

**Figure 9.** Comparison of the conductivities and activation energies as a function of the free lattice volume for the low-temperature (T < 500 °C) region in wet air for the samples: $La_2ScZnO_{5.5}$ (LSZ), $La_2Sc_{0.9}Mg_{0.1}ZnO_{5.45}$ (LZSMg), $La_2Sc_{0.9}Ca_{0.1}ZnO_{5.45}$ (LZSCa), and $La_{1.9}Ca_{0.1}ScZnO_{5.45}$ (LCaZS).

Since the concentration of protons is approximately equal (according to the TG data), an increase in proton conductivity is associated with an increase in the mobility of protons. This trend, shown in Figure 9, reflects the fundamental relationship between the

dynamics of oxygen sublattice and proton mobility [60]. There are many examples in the literature to support this assumption. For example, Iwahara et al. showed that the protonic conductivities were lower for $CaZrO_3$-doped ceramics than those for $SrZrO_3$ and $BaZrO_3$-based materials; furthermore, the proton conductivity increased with an increasing ionic radius of the B-site cation dopant Ga-In-Y in $BaZr_{0.95}M_{0.05}O_{3-\delta}$ [61]. Haugsrud et al. [62] explained the decrease in protonic conductivity for Ca-doped rare-earth niobates $Re_{1-x}Ca_xNbO_4$ (Re = La, Nd, Gd, Tb, Er, Y) with a decreasing radius of the rare-earth cations (La >Nd > Tb > Er) by the decrease in polarizability of the oxide sublattice. In other words, if the distance between the oxygen ions is greater, then the lattice becomes more flexible and proton transport becomes faster; as a result, the proton conductivity increases.

It should be noted that the enhancements of the bulk conductivity in wet air in comparison with dry air at temperatures below 400 °C reached ~0.5 orders of magnitude for the composition $La_2Sc_{0.9}Ca_{0.1}ZnO_{5.45}$. For the undoped phase and for the sample $La_2Sc_{0.9}Mg_{0.1}ZnO_{5.45}$, these differences in conductivities in wet and dry air were not significant, although the phases are characterized by the presence of proton current carriers. Such behavior can be explained by the presence of holes in dry air. If the atmosphere is oxidizing (air), a formation of electronic holes will follow in order to compensate for the oxygen vacancies according to the equation:

$$V_o^{\bullet\bullet} + \frac{1}{2}O_2 \Leftrightarrow O_o^{\times} + 2h^{\bullet}, \tag{10}$$

where $V_o^{\bullet\bullet}$ is oxygen vacancy, $O_o^{\times}$ are regular oxygen sites, $h^{\bullet}$ is a hole.

In wet air, the concentration of holes decreases as a result of interacting with water vapor according to the equation:

$$H_2O + 2h^{\bullet} + 2O_o^{\times} \Leftrightarrow \frac{1}{2}O_2 + 2(OH)_o^{\bullet}, \tag{11}$$

where $O_o^{\times}$ are regular oxygen sites, $h^{\bullet}$ is a hole, and $(OH)_o^{\bullet}$ are defects that appear when protons are localized at regular oxygen sites. Thus, the conductivity behavior of $La_2ScZnO_{5.5}$ and $La_2Sc_{0.9}Mg_{0.1}ZnO_{5.45}$ samples in a wet atmosphere indicates a complex combination of proton and hole conductivities. A decrease in the concentration of most mobile hole carriers leads to a decrease in conductivity; however, with a decrease in temperature, a gradual increase in the concentration of proton carriers can compensate for this effect and even increase the total conductivity.

Nevertheless, to confirm this assumption, it is necessary to carry out additional studies on the differentiation of the conductivity into ion and electrical contributions.

A comparison of the conductivity $La_2ScZnO_{5.5}$ with the $Ca^{2+}$-doped sample based on $LaScO_3$ with the composition $La_{0.9}Ca_{0.1}ScO_{3-\delta}$ [33], is also shown in Figure 7. As can be seen, in wet air, the doped sample $La_{0.9}Ca_{0.1}ScO_{3-\delta}$ exhibited high conductivity, which exceeded by 0.5 orders of magnitude that of $Ca^{2+}$-doped $La_2ScZnO_{5.5}$. Since the concentrations of protons in the doped compound $La_{0.9}Ca_{0.1}ScO_{3-\delta}$ were small [33], the high conductivity in wet air was due to the high mobility of protons in comparison with the investigated doped phases based on $La_2ScZnO_{5.5}$. It can be assumed that the protons in the $Ca^{2+}$-doped $La_2ScZnO_{5.5}$ compounds are more trapped than in doped $LaScO_3$ and are less mobile.

Although the obtained phases are inferior in terms of conductivity, the introduction of $Zn^{2+}$ makes it possible to significantly reduce the sintering temperature of the material. Thus, dense ceramics based on perovskite $LaScO_3$ are obtained at 1650 °C [33], and $Zn^{2+}$-containing phases have good ceramics when sintered at 1300 °C.

These studies have shown that the phase $La_2ScZnO_{5.5}$ is promising for further investigations, both when varying the concentration of the dopants, and when the nature of the latter changes. Probably, the selection of the optimal concentration of the dopant with its optimal size will increase the ionic conductivity. From this point of view, the new phases

capable of low-temperature sintering, having chemical stability and high ionic conductivity could be used as electrolytes for fuel cells.

## 4. Conclusions

A novel perovskite $La_2ScZnO_{5.5}$ and doped phases $La_2Sc_{0.9}Mg_{0.1}ZnO_{5.45}$, $La_2Sc_{0.9}Ca_{0.1}ZnO_{5.45}$, and $La_{1.9}Ca_{0.1}ScZnO_{5.45}$ were obtained. An analysis of the effect of doping on the electrical properties was carried out. It was established that the introduction of small dopant concentrations leads to an increase in conductivity due to an increase in oxygen vacancies concentration. The conductivities increased in the order of $La_2ScZnO_{5.5} < La_2Sc_{0.9}Mg_{0.1}ZnO_{5.45} < La_2Sc_{0.9}Ca_{0.1}ZnO_{5.45} < La_{1.9}Ca_{0.1}ScZnO_{5.45}$ with an increase in the free volume of migration. All of the phases are capable of incorporating water. The proton concentration increased in the order: $La_2ScZnO_{5.5} < La_2Sc_{0.9}Mg_{0.1}ZnO_{5.45} < La_2Sc_{0.9}Ca_{0.1}ZnO_{5.45} \approx La_{1.9}Ca_{0.1}ScZnO_{5.45}$. It was established that for the samples $La_2ScZnO_{5.5}$ and $La_2Sc_{0.9}Mg_{0.1}ZnO_{5.45}$, full filling of oxygen vacancies by water was not reached, but the samples $La_2Sc_{0.9}Ca_{0.1}ZnO_{5.45}$ and $La_{1.9}Ca_{0.1}ScZnO_{5.45}$ exhibited a ~90% hydration limit, i.e., almost filling the concentration of the oxygen vacancy. It was found that the conductivities in wet air increased in the same order: $La_2ScZnO_{5.5} < La_2Sc_{0.9}Mg_{0.1}ZnO_{5.45} < La_2Sc_{0.9}Ca_{0.1}ZnO_{5.45} < La_{1.9}Ca_{0.1}ScZnO_{5.45}$.

Thus, the method of creating novel oxygen-deficient perovskite $La_2ScZnO_{5.5}$ was successfully performed. The introduction of zinc makes it possible to obtain high-density ceramics while maintaining a significant level of ionic ($O^{2-}$, $H^+$) conductivity, which is an important requirement for fuel cells. Moreover, reducing the scandium content by the introduction of zinc will significantly reduce the cost of the materials for solid electrolytes.

**Author Contributions:** Conceptualization, I.A. and K.B.; methodology, I.A.; investigation, S.P. and K.B.; data curation, K.B. and A.E.; writing—original draft preparation, K.B. and I.A.; writing—review and editing, K.B. and I.A. All authors have read and agreed to the published version of the manuscript.

**Funding:** This research received no external funding.

**Institutional Review Board Statement:** Not applicable.

**Informed Consent Statement:** Not applicable.

**Acknowledgments:** The study was financially supported by the Ministry of Education and Science of the Russian Federation (state assignment no. AAAA-A20-120061990010-7).

**Conflicts of Interest:** The authors declare no conflict of interest.

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
