# Peer review of "Crystal Structure, Electrical Conductivity and Hydration of the Novel Oxygen-Deficient Perovskite La2ScZnO5.5, Doped with MgO and CaO"

_applsci, doi:10.3390/app12031181_

Round 1
Reviewer 1 Report
The paper presents a work regarding a novel oxygen-deficient perovskite La2SсZnO5.5 doped with MgO and CaO. The aythors analyzed the effect of doping on the electrical properties of the obtained compounds.
The focus of the paper is interesting. In my opinion the manuscript may be acceptable for publication in the Applied Sciences after minor revision.
My specific comments:
The Materials and Methods section requires additional information:
- Dimensions of the tested samples.
- The number of tested samples / number of tests is not given. What is the repeatability of the test results, what is the error in the results, are the obtained results representative?
- The authors did not provide information on the parameters of impedance measurements: frequency range, amplitude of the perturbation signal, auxiliary electrodes.
Figure 3b):
There is no explanation of what the two graphs represent - drawn with filled and unfilled circles. There is no description of the electrical equivalent circuit given in this figure.
Reviewer 2 Report
The manuscript titled, “Crystal structure, electrical conductivity, and hydration of the novel oxygen-deficient perovskite La2ScZnO5.5, doped with MgO and CaO” by Belova et al did a detailed study on engineering the novel doping on perovskite-based material. The present manuscript is interesting and the author’s presentation and results part are good. However, authors must address the following minor issues in their manuscript before being published in Applied Science.
Minor revision
- In XRD patterns, the peak results for all the prepared samples irrespective of doping remain the same. An author should explain the variations or shifts and compare to metal oxide in detail in the revised manuscript (10.1016/j.envres.2021.112050).
- The author should explain the better withstanding property of the un-doped material compared to the doping components.
- In Nyquist plots, the author should explain the curves from 800 ℃ to 700 ℃.
- The author should briefly state the change in temperature of two measurements (Figure 3a and 3b) in Nyquist plots for different partial pressures.
- Some of the important references need to discuss on TG analysis and cite in the revised Results and Discussion part (10.1007/s11581-013-0985-z).
- An author should check for grammatical errors and details in some parts of the manuscript.
Reviewer 3 Report
Reviewed paper “Crystal structure, electrical conductivity and hydration of the novel oxygen-deficient perovskite La2SсZnO5.5, doped with MgO and CaO” deals with an interesting topic and can be interesting for readers of Applied Sciences journal. This manuscript seems to provide interesting insights on a topic of perovskite materials, so it may be of interest to the materials community.
Authors presents information about novel oxygen-deficient perovskite La2SсZnO5.5 and doped phases La1.9Ca0.1ScZnO5.45, La2Sc0.9Ca0.1ZnO5.45, La2Sc0.9Mg0.1ZnO5.45 obtained by a solid-state reaction process. XRD and ТGА+MS techniques to obtain phase composition was conducted. Authors measured materials' conductivity by electrochemical impedance technique under the atmosphere of various water vapor partial pressures.
This paper suits the requirements of the journal. The paper contains 7 figures, 1 table and 10 formulas – most figures are legible and good quality.
English of the paper is rather poor (not good) – in my opinion the language of the paper should be a little improved. I am asking for corrections by a native speaker.
Primary criteria:
- Does this manuscript offer original data and innovative insights?
Yes, the results presented appear to be original and potentially innovative.
- Does the subject matter manuscript have wide scientific interest and potential applicability?
The subject matter is relevant to the perovskite materials community and is potentially applicable for example in fuel cells.
- Have the authors remembered to cite previous seminal papers on the subject?
The authors have cited 51 literature items – the numbers is not sufficient and not all items are significant on the subject.
- Is the Supplementary Material (Supporting Information) helpful, appropriate and error-free?
N/A
I find some mistakes for example:
- The paper should by reformatted following IMRAD structure. It would be useful to add a separate section called Experimental. Please use the names of the sections in order: Introduction, Materials and Methods, Experimental, Results and Discussion, Conclusions.
- Abstract should be thoroughly improved. Please present in the abstract in a synthetic way the subject of the article in the following: introduction, assumptions for the paper, experimental (research) methods, obtained research results (novelty) and conclusions.
- Introduction section should be correct. Authors should include new information about topic of a paper. More information based on worldwide (global) study. The list of references should also be changed.
- Experimental – please describe all equipment and/or software used in the experiment – please insert model of equipment (manufacturer, city, country).
- Please describe the experimental procedure which was carried out in a simple and clear way, for example by means of flowcharts.
- In my opinion Conclusions chapter should be a little changed. It should contain the most relevant information contained in Results and Discussion
- In the final part of the Results and Discussion section, please write about the potential use (application) of this type of material.
- References section is sufficient but papers cited in the references 43 from all 51 are older then 5 years – these publications constitute 84 % of all cited papers. I propose to add some new (from the last 5 years). Author should include several modern papers of global research in this field.
- In the list of references I found 13 paper of the Authors of reviewed paper. Please indicate the differences in the studies presented in the cited article and their relation to the presented topic.
- Please describe all the variables found in the formulas.
- In the whole paper (text and tables), you write for example 8.33% (for example line: 56) – you should write this value and symbol with a space as “8.33 %”.
- Please prepare a literature review and body text of a paper according to the guidelines of the Applied Sciences journal and MDPI Publisher.
The results obtained are interesting and promising but the manuscript can be accepted for publication in Applied Sciences journal after MAJOR corrections (Reconsider after major revision).
Round 2
Reviewer 3 Report
In corrected paper “Crystal structure, electrical conductivity and hydration of the novel oxygen-deficient perovskite La2SсZnO5.5, doped with MgO and CaO” Author has properly addressed the concerns from the referee. All my remarks have been included in the revised document. Below you will find my comments on the attached answers.
Referring to my substantive reservations – the authors made the necessary modifications. They changed the text of the article and removed stylistic and grammatical errors.
Author reformatted and extended Abstract, Introduction, Materials and Methods chapter and body text of a whole paper. Author significantly reformat the entire article. They modified a list of a references by including the most recent relevant publications published in last years.
The manuscript can be ACCEPTED for publication in Applied Sciences journal in the current form.